# Cellular and molecular roles of reactive oxygen species in wound healing
Matthew Hunt[1], Monica Torres[1,2], Etty Bachar-Wikstrom [1] & Jakob D. Wikstrom [1,2] ✉

Wound healing is a highly coordinated spatiotemporal sequence of events involving several cell types and tissues. The process of wound healing requires strict regulation, and its disruption can lead to the formation of chronic wounds, which can have a significant impact on an individual's health as well as on worldwide healthcare expenditure. One essential aspect within the cellular and molecular regulation of wound healing pathogenesis is that of reactive oxygen species (ROS) and oxidative stress. Wounding significantly elevates levels of ROS, and an array of various reactive species are involved in modulating the wound healing process, such as through antimicrobial activities and signal transduction. However, as in many pathologies, ROS play an antagonistic pleiotropic role in wound healing, and can be a pathogenic factor in the formation of chronic wounds. Whilst advances in targeting ROS and oxidative stress have led to the development of novel pre-clinical therapeutic methods, due to the complex nature of ROS in wound healing, gaps in knowledge remain concerning the specific cellular and molecular functions of ROS in wound healing. In this review, we highlight current knowledge of these functions, and discuss the potential future direction of new studies, and how these pathways may be targeted in future pre-clinical studies.

Dermatological would healing is the tightly coordinated response of restoring skin tissue integrity and homeostasis following damage. Involving numerous immune and non-immune cell types as well as associated cytokines, growth factors, and extracellular components – in a healthy, acute response – orderly wound healing consists of the four consecutive stages of haemostasis, inflammation, proliferation, and tissue remodelling[1,2]. Briefly, during the haemostasis phase, platelets are recruited or leak out of damaged vasculature and with the simultaneous activation of the coagulation cascade and formation of fibrin fibres, form a clot at the wound site[3]. Next, during inflammation, various immune cells migrate to the wound site and utilise phagocytotic effects to protect against infection. Additionally, these immune cells also release pro-inflammatory cytokines and growth factors to induce the activation of fibroblasts, keratinocytes, and endothelial cells, as well as prepare the wound bed for the formation of granulation tissue[4,5]. During the proliferation phase, granulation tissue is formed and damaged tissue is replaced. At the remodelling phase, connective tissue, replacement epithelium, and scar tissue are all formed[6].

Reactive oxygen species (ROS) play an essential pleiotropic role in wound healing, and several ROS are involved in the wound healing milieu (Fig. 1). These include superoxide ($O_2^-$), hydroxyl radicals (.OH) and ions ($OH^-$), hydrogen peroxide ($H_2O_2$), and peroxide ($.O_2^{-2}$)[7]. ROS are implicated in numerous pathophysiological functions within the wound healing

process such as anti-bacterial activities[8,9], as well as acting as secondary messengers in signalling cascades to modulate chemotaxis, angiogenesis, cell growth and migration, stem cell fate, and extracellular matrix (ECM) deposition[7,10–13].

Importantly, both the levels and timing of ROS production need to be tightly regulated for efficient wound healing[7]. Too high levels caused by either excess ROS production or impaired detoxification lead to oxidative stress, elevated tissue damage, and pathophysiological stalling[14,15], whilst too low levels impede cellular and molecular processes of wound healing which are dependent on ROS-mediated signal transduction[7] – ultimately leading to the formation of chronic wounds. Highlighting the delicate and complex balance required, inhibition of ROS has been shown to impair wound healing in numerous animal models[16–23], whilst improvements in anti-oxidant capabilities have been shown to be beneficial in treatments of chronic and diabetic wounds[14,24,25]. As such, due to the multifaceted nature of chronic wound pathogenesis and susceptibility to abnormalities in ROS balance, interest in the role of ROS in wound healing, as well as the potential applicability of targeting ROS therapeutically, has grown significantly in recent years[14]. However, it will be essential to further elucidate the precise signalling pathways and mechanisms in which ROS is involved in wound healing. Thus, in this review, we discuss the current knowledge of the cellular and molecular roles of ROS in wound healing and chronic wound

[1]Dermatology and Venereology Division, Department of Medicine (Solna), Karolinska Institutet, Stockholm, Sweden. [2]Dermato-Venereology Clinic, Karolinska University Hospital, Stockholm, Sweden. ✉e-mail: jakob.wikstrom@ki.se

## Normal wound healing

**Fig. 1 | Summary of ROS activities during the wound healing process.** Stages of wound healing with illustrations of the various beneficial roles that physiological levels ROS play in the respective stages, in addition to the roles excessive ROS and oxidative stress play in chronic wound pathogenesis. During the haemostasis stage, NO prevents platelet adhesion to vessel walls, whilst ROS such as $O_2^-$ increases fibrin deposition, and $H_2O_2$ induces the recruitment of monocytes and neutrophils. During inflammation, ROS play important roles in activating immune cells, as well as eliminating pathogens and preventing infection. During the proliferation stage,

ROS play vital roles in modulating numerous cellular signalling pathways to promote the proliferation, migration, and differentiation of fibroblasts and keratinocytes, as well as angiogenesis, ultimately promoting collagen remodelling and extracellular matrix formation. Oxidative stress caused by excessive levels of ROS contribute to the pathogenesis of chronic wounds in various ways, including by increasing apoptosis, promoting pathogen expansion and thus infection, as well as impairing the correct modulation of cell signalling pathways involved in cell dynamics.

pathogenesis, as well as evaluate recent advances in pre-clinical therapeutic approaches targeting ROS and oxidative stress.

## Physiological functions of ROS

ROS encompass both free-radical or non-radical derivative (peroxides) oxygen intermediates generated by plasma membrane proteins[26] (Fig. 2). Physiologically, as well as in pathologies such as wound healing, $H_2O_2$ is recognised as the predominant paracrine ROS secondary messenger involved in signalling cascades[27,28]. This is due to the fact that $H_2O_2$ can quickly and readily diffuse through cell membranes, primarily through aquaporins (AQPs)[29,30], as well as between neighbouring cells through gap junctions – hemichannels composed of connexins which facilitate the transfer of molecules 1–3 kDa large such as ROS between cells to propagate oxidative signals[31]. Additionally, ROS can directly modulate post-transcriptional gene regulation by interacting and reversibly oxidising thiolate groups and methionine[32], as well as activating mitosis-related signal transduction pathways[8,33,34] and electron-rich cysteine residues[35].

The main sources of intracellular $H_2O_2$ are NADPH oxidases (NOXs) and dual-oxidases (DUOXs)[36–38], in conjunction with superoxide dismutases (SOD), as well as at the mitochondrial electron transport chain (ETC)[39,40] – highlighting an important aspect of mitochondria within wound healing[41]. Collectively, NOXs and the ETC generate roughly 85% of $H_2O_2$, with the remaining production deriving from oxidases in the endoplasmic reticulum (ER) and peroxisomes, as well as from cumulative environmental stressors such as UV or ionising radiation[8,40,42,43]. Additionally, membrane-bound NOXs are also responsible for producing $.O_2^-$ utilised in antimicrobial activities[44]. Other ROS are produced by cytosolic enzymes such as cyclooxygenase[45], or during lipid metabolism within peroxisomes[46].

As previously mentioned, whilst low to moderate physiological levels of ROS are beneficial for several processes of wound healing pathophysiology, excess ROS can be deleterious. To counteract these harmful effects, a variety of antioxidant enzymes play vital roles in maintaining ROS levels, termed redox balance. These include peroxiredoxins[47] such as catalase (CAT)[48], glutathione peroxidases[49,50], and mitochondrial nicotinamide nucleotide

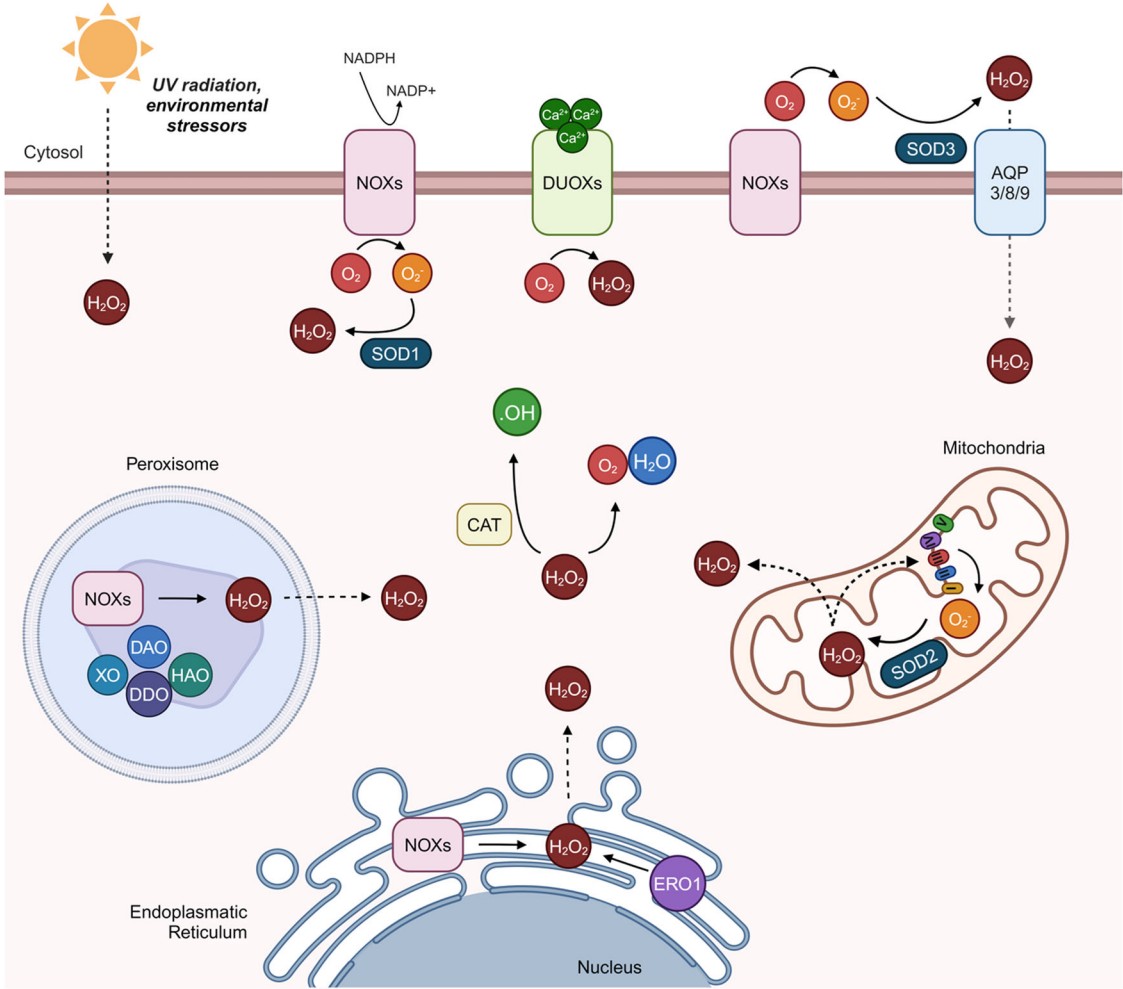

**Fig. 2 | Cellular ROS homeostasis.** Schematic diagram depicting the various ROS-generating pathways occurring within cells. At the cell membrane, $O_2^{.-}$ is converted from $O_2$ in an NADPH-mediated reaction by NOXs, which is then converted to $H_2O_2$ by SOD1. $H_2O_2$ can also be produced from $O_2$ in a $Ca^{2+}$-mediated reaction by DUOXs or by UV radiation or other environmental stressors. Extracellular $H_2O_2$ can also be imported into cells through AQPs 3, 8, or 9. Within cells, $O_2^{.-}$ leaks from the ETC during oxidative phosphorylation (OXPHOS) and is converted into $H_2O_2$ by SOD2/MnSOD2 and effluxed out of mitochondria into the cell cytosol.

Additionally, $H_2O_2$ can be produced in the ER by either ERO1 or NOXs and effluxed into the cytosol, as can $H_2O_2$ produced by NOXs within peroxisomes. XO, DAO, DDO, and HAO are also produced in peroxisomes. Within the cell cytoplasm, $H_2O_2$ can be detoxified into $H_2O$ and $O_2$ as well as .OH by CAT. ER endoplasmic reticulum, SOD superoxide dismutase, AQP aquaporin, CAT catalase, XO xanthine oxidase, DAO D-amino acid oxidase, HAO 2-hydroxy acid oxidase, ERO ER oxidoreductin.

transhydrogenase (NNT)[51], which act as 'sinks' to remove $H_2O_2$ and maintain non-deleterious physiological levels. In addition, SOD, of which there are four isoforms in humans, are antioxidant metalloproteinases which regulate levels of $O_2^{.-}$. In particular, the mitochondrial SOD (MnSOD/SOD2) converts .$O_2^{.-}$ into $H_2O_2$, which is less reactive than .$O_2^{.-}$ and so can readily be used for cellular signalling. As SOD2 is induced by hypoxia and subsequent HIF-1α activation, it is highly upregulated after wounding[52,53].

Oxidative stress is the state induced by an overbalance in the form of excess ROS, and can be a causative factor in chronic wound formation[15], as well as in the pathogenesis of other diseases including cancers, cardiovascular diseases, Parkinson's, obesity, and other clinically-relevant age-related diseases[54–57]. In the context of wound healing, oxidative stress can lead to the existence of a prolonged pro-inflammatory environment as well as dysregulated re-epithelialisation, as discussed in the following sections[58].

Finally, another important free radical is nitric oxide (NO), which is a reactive nitrogen species (RNS) involved in vascularisation, inflammation, and antimicrobial activities in wound healing[59–61]. Most importantly with regards to wound healing, NO plays an important role in pathogen

clearance during the inflammatory stage, and this NO is produced by the inducible nitric oxide synthase (iNOS) isozyme[62,63]. Here, NO targets both gram-negative and -positive bacteria through aberrant peroxidation and the production of ONOO⁻, although this can be hindered by its short half-life[64,65]. Alternatively, lower levels of NO, produced by endothelial nitric oxide synthase (eNOS), play important roles in preventing platelet adhesion to vessel walls during haemostasis[66], and for both keratinocyte and fibroblast proliferation, motility, and differentiation at the later proliferation and tissue remodelling stages of wound healing[67–69]. Insufficient production of NO has been shown to be a significant factor in the development of chronic wounds such as diabetic foot ulcer (DFU), primarily due to resultant impaired antimicrobial activities[63,70].

## ROS and immune cell function during wound healing
### ROS and leucocyte recruitment
Immediately following skin wounding there is a peak in ROS production to ~0.5–50 µM. Here, ROS are utilised to simultaneously recruit leucocytes to the wound site, as well as to induce vasoconstriction[16,17,71]. Through studies of ROS dynamics in embryonic zebrafish wounding, Niethammer et al.

demonstrated for the first time that epithelial cell production of $H_2O_2$ preceded the recruitment of leucocytes, and in particular that DUOX was the main source of $H_2O_2$ at the wound site and inducer of rapid leucocyte recruitment from long distances[17]. In embryonic *Drosophila*, the activation of DUOX and subsequent $H_2O_2$ production was shown to be triggered by wound-induced calcium ($Ca^{2+}$) flashes, where $Ca^{2+}$ binds to an EF hand $Ca^{2+}$-binding motif of DUOX[72].

Expanding on this work, Yoo et al. demonstrated the cystine residue C466 on the Src family kinase Lyn as being the direct target of $H_2O_2$ to induce neutrophil recruitment to the wound site, mediated through ERK signalling[16]. Alternatively, the activation of DUOXs, with subsequent $H_2O_2$ production and neutrophil recruitment, can also be activated by ATP through the P2Y receptor (P2YR)/phospholipase C (PLC) $Ca^{2+}$ signalling pathway following wounding in embryonic zebrafish tailfins[73].

## ROS and macrophage function

Macrophages play important roles within the wound healing process, including in antimicrobial activities, as well as inflammation, angiogenesis, anti-inflammation, re-epithelialisation, and tissue resolution[74]. ROS-induced HIF-1α stabilisation leads to the activation of macrophages in the early stages of wound healing, and promotes metabolic reprogramming towards glycolysis[75,76], as well as increased angiogenesis[76]. Importantly, both NOX1- and NOX2-produced ROS are required for the activation and differentiation of monocytes into proinflammatory M1 and anti-inflammatory M2 macrophages[77]. Additionally, in atherosclerotic lesions, NOX4-produced ROS drives monocyte and macrophage cell death[77] – a vital step required to prevent prolonged inflammation in wound healing[78].

Although NOXs are essential for macrophage activation, they have been shown to be dispensable for M1 macrophage-mediated proinflammatory cytokine production[79]. Instead, pro-inflammatory cytokine production and inflammasome activation in macrophages predominantly relies on the regulation of the Nrf2 response[80], which can be primarily activated by glutathione or thioredoxin systems, as well as to a lesser extent NOXs[81]. Other important pro-inflammatory signalling pathways regulated by ROS – and in particular $H_2O_2$ – include p38-MAPK-mediated NF-κB/HIF-1α[82,83], and JAK-STAT pathways[84].

Monoamine oxidases (MAOs) – a mitochondrially-located enzyme responsible for catalysing the oxidative deamination of $H_2O_2$[85] – is upregulated by the M2 macrophage-activating IL-13 and IL-4, or LPS signals. This process is mediated through JAK signalling pathways and is thus important in anti-inflammation and re-epithelialisation during wound healing[86]. Indeed, MAO inhibitors significantly reduced $H_2O_2$ levels and NF-κB/TNFα activation to impair apical migration and proliferation of junction epithelium in a rat chronic wound model[87]. Alternatively, DUOX-induced ROS stimulated the activation of macrophages and promoted epithelial proliferation in a JNK-dependent manner during *Drosophila* epithelial disc healing[88].

Of note, whilst many studies have demonstrated the importance of ROS and macrophage function in other pathologies and physiological contexts, few have specifically investigated this link within wound healing pathophysiology[89]. Importantly, the majority of studies in this area used the oversimplistic M1 and M2 macrophage classifications, whereas in recent years advancements have been made to study the more elaborate classifications of macrophages in the general immunology setting[90], and this should thus be applied to the macrophages in wound healing setting[91]. Single-cell RNA-sequencing (scRNAseq) and other advanced omics-based techniques have in recent years significantly powered the investigation of the roles of specific cell lineages, including macrophage lineages, in different stages of wound healing[76,92]. As such, future investigations seeking to elaborate on the roles of ROS dynamics in wound healing would greatly benefit from the utilisation of these techniques.

## ROS and antimicrobial activities

One of the major factors that leads to the formation of chronic wounds is that of prolonged inflammation and pathogenic infection, which can hamper angiogenesis, stem cell function, and extracellular matrix remodelling[93]. During acute wound healing, immune cells are required to eliminate pathogens and prevent infection at the wound site, and ROS play an essential role in this function[94] (Fig. 3). Here, extracellular

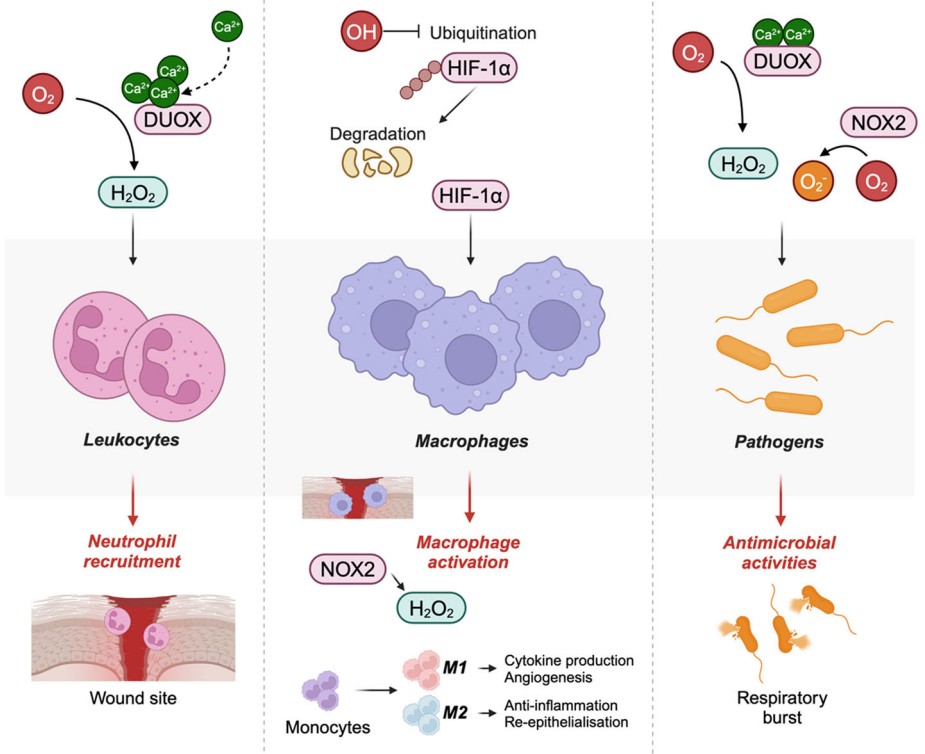

**Fig. 3 | Roles of ROS utilisation in immune cells during wound healing.** Schematic diagram depicting the roles of ROS in neutrophil and macrophage recruitment, as well as in antimicrobial activities during the inflammatory stage of wound healing. Here, wounding-induced $Ca^{2+}$ flashes lead to the upregulation of DUOX-mediated $H_2O_2$ production, stimulating the recruitment of leukocytes to the wound site. Additionally, OH prevents the ubiquitination and subsequent degradation of HIF-1α, leading to increased HIF-1α signalling and macrophage activation, primarily mediated through $H_2O_2$ signalling. Finally, immune cells utilise various reactive oxygen species, including $O_2^-$, to destroy pathogens through respiratory bursts.

$H_2O_2$ – generated by DUOXs in response to the increased $Ca^{2+}$ binding following wounding – promotes the production of bacteria-destroying ROS after reacting with halide or thiocyanate[95,96]. In the absence of this function, polymicrobial biofilms form, which can lead to the expansion of a pathogenic environment and ultimately the stalling of the wound healing response and thus chronic wound formation[97].

In a dual action, peroxidases such as myeloperoxidase and eosinophil peroxidase convert $H_2O_2$ into other oxidants such as hypochlorous acid, which is then used by neutrophils in antibacterial activities[98,99]. This process additionally prevents the toxic build-up of $H_2O_2$ which can occur in the early stage of wound healing following leucocyte recruitment, where $H_2O_2$ levels are highest[100,101].

Macrophages also play an important role in clearing pathogens during wound healing through phagocytosis. Here, NOX2-derived $O_2^{.-}$ is released into phagocytotic vesicles to kill internalised pathogens through respiratory bursts[89].

## Role of ROS in re-epithelialisation
### ROS in platelet aggregation and angiogenesis
Moderate amounts of ROS (up to a 40% increase) are required for the reduction in platelet adhesion to collagen surfaces and thus platelet activation[102,103]. In light of this, and conversely, whilst ROS is known to accelerate platelet function in wound healing, transfer of platelet-derived mitochondria into diabetic mice improved wound healing in part by preventing the overexpression of ROS[104]. Importantly, $H_2O_2$ induces the recruitment of vascular smooth muscle cells to the wound site[11,105].

As previously mentioned, NO plays an important role in angiogenesis during wound healing. Here, elevated NO production – as a result of increased activation of NOXs, in particular NOX4 – leads to the stabilisation of HIF-1α and thus promotion of endothelial cell (EC) survival, migration, differentiation, and therefore neovascularisation[64,106–108]. Highlighting this, near-infrared (NIR)-triggered NO production supressed the proteasomal degradation of HIF-1α. Here, by preventing the interaction of HIF1-α with E3 ubiquitin ligases, both VEGF and CD31 expression was enhanced in ECs, coinciding with increased cell proliferation and migration – collectively accelerating wound healing in diabetic mice[64]. $H_2O_2$ produced by NOX4 also activated both the TRPM2[109], and SERCA2 channels[110] to promote $Ca^{2+}$ uptake and thus improve EC activity (Fig. 4). In wound healing and other hypoxic-state pathologies such as brain ischemia, these ROS-derived effects on ECs are associated with phosphorylation-dependent activation of various signalling molecules including those of ERK, c-JUN, MAPK, AKT, SMAD, and JNK[111].

Finally, production of $O_2^{.-}$ by both NOX2 and NOX4 also leads to the upregulation of VEGF[110,112]. In particular, NOX2 was demonstrated to stimulate VEGFR2 and angiogenesis in wounds through the activation of NF-κB by 2-deoxy-D-ribofuranose 1-phosphate (dRP) – an intermediate of pyrimidine metabolism. This NOX2-derived ROS was primarily generated by both platelets and macrophages[13,113].

### $H_2O_2$ mediated cell signalling and re-epithelialisation
Many cytoskeletal proteins possess cysteines which are highly sensitive to oxidation[114], and in a complementary fashion, production of $H_2O_2$ occurs primarily in leading edge cells involved in re-epithelialisation[115]. In particular, $H_2O_2$ promotes actin cytoskeleton reorganisation and cell migration by directly oxidising actin and actin-binding proteins[116], as well as activating numerous cell signalling pathways associated with re-epithelialisation.

As discussed previously, there are several mechanisms in which ROS-mediated signalling cascades are initiated in wound healing (Fig. 5). Indeed, ROS production required for both immune cell function and re-epithelialisation share similar stimuli. In one key example, Hunter et al. demonstrated in *Drosophila* embryo healing that mitochondrially-derived $H_2O_2$ – produced downstream of intracellular $Ca^{2+}$ bursts – led to the polarisation of the actomyosin cytoskeleton and E-cadherin distribution around the wound to promote wound healing[117]. Specifically, this action occurred via oxidation of the Src kinase Src42, and supported results from a previous study in which mitochondrial ROS (mtROS) was produced downstream of $Ca^{2+}$ bursts following wounding[118]. In addition to $Ca^{2+}$-mediated activation, DUOX can also be activated downstream of extracellular ATP-activated purinergic receptors[119–121].

One cell signalling pathway regulated by wounding-induced ROS is that of c-JUN. Here, the inhibition of wounding-induced ROS accumulation significantly inhibited healing in planarian worms by preventing F-actin reorganisation and epithelial cell rearrangements – mediated through c-JUN activation at the wound site[122]. Separately, ROS-mediated activation of c-JUN also accelerated wound healing in diabetic rats through increases in angiogenesis and re-epithelialisation[123], whilst NOX-produced $H_2O_2$-activation of the JNK pathway increased epithelial cell proliferation in adult zebrafish tailfin healing[19].

Alternatively, $H_2O_2$ also regulates MAPK signalling during wound healing – namely, through thioredoxin (Trx) oxidation and PI3K/AKT1-Ask1-MAP3K-mediated activation of JNK and p38[22,26,124]. Interestingly, in a study investigating the interplay between ROS and AKT signalling in *Drosophila* regeneration, the importance of nutrient sensing and metabolism in this pathway was highlighted[125]. Here, ROS-mediated

**Fig. 4 | ROS and endothelial cell function.**
Schematic depicting how $H_2O_2$ produced by NOX4 increases $Ca^{2+}$ uptake into endothelial cells through elevated SERCA2 and TRPM2 channel activity, subsequently leading to increased endothelial cell division and migration – thereby promoting angiogenesis.

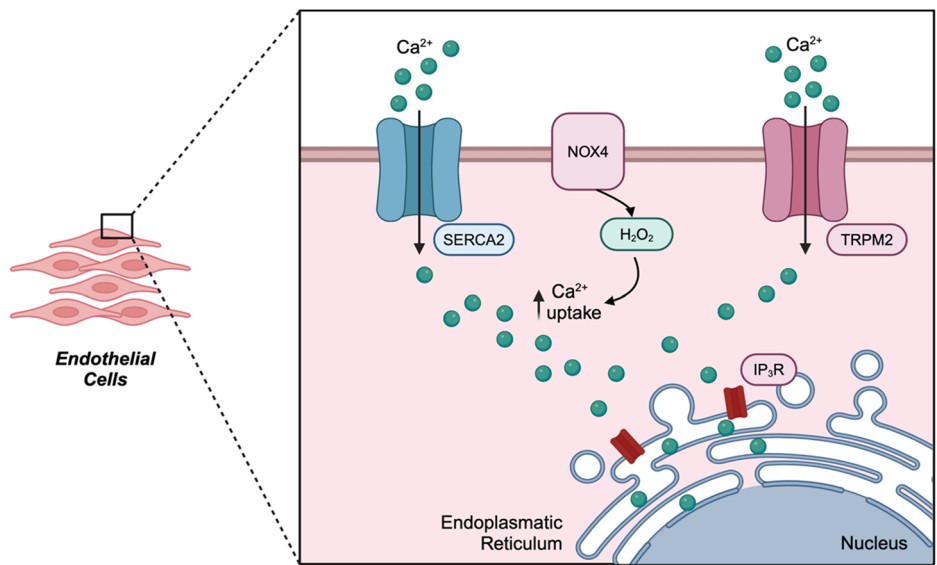

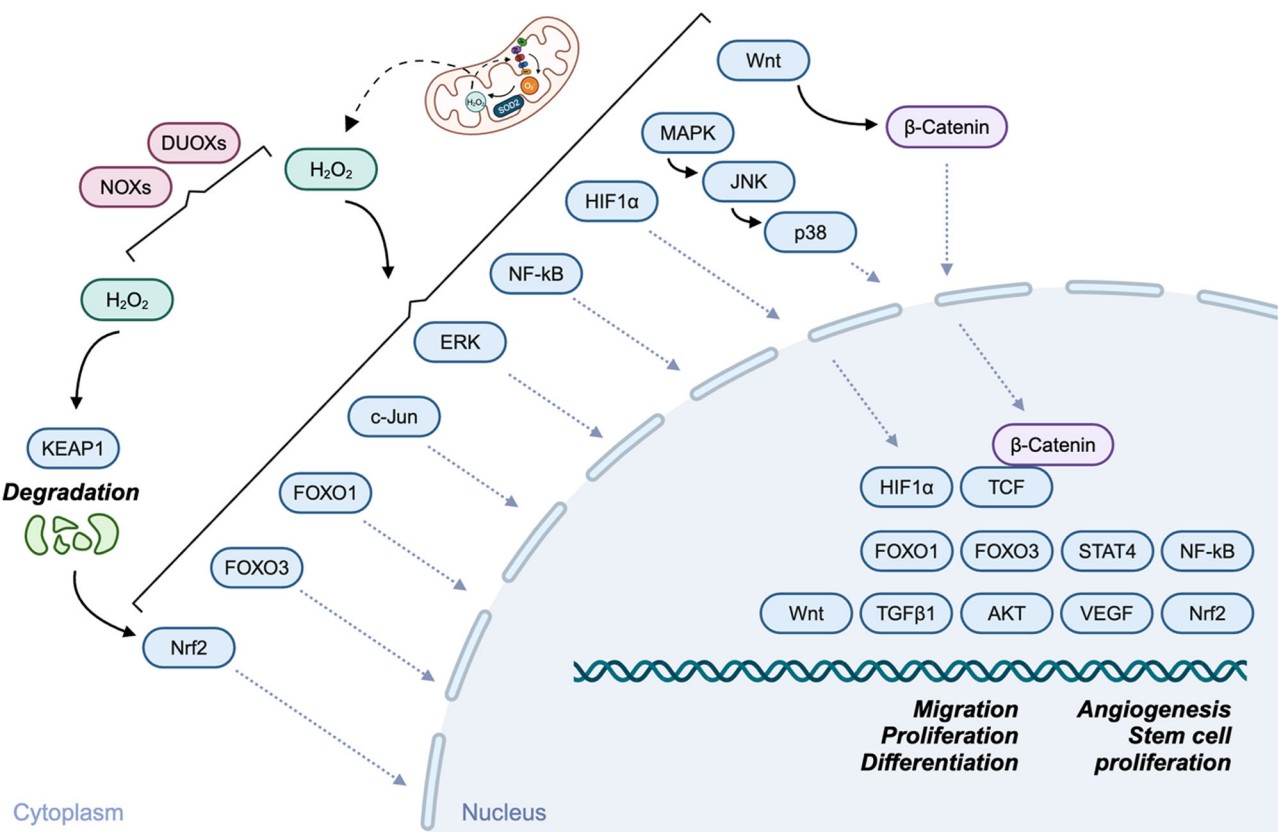

**Fig. 5 | ROS and re-epithelialisation.** Schematic showing the various signalling molecules and pathways which are modulated by ROS to regulate re-epithelialisation during wound healing. $H_2O_2$ produced by either NOXs or DUOXs, or derived from mitochondria, stimulate numerous cell signalling pathways which ultimately lead to the upregulation of processes to accelerate wound healing – such as cell migration, proliferation, differentiation, angiogenesis, or stem cell propagation.

phosphorylation of Ask1 at Ser38 induced p38-mediated regeneration in a nutrient-sensitive insulin signalling manner, supporting similar findings in studies of stress-induced regeneration in the gut[126]. As metabolic regulation of fibroblasts and keratinocytes during re-epithelialisation is known to be important[127], future studies should aim to further investigate the role of the MAPK and other relevant signalling pathways.

Another pathway that $H_2O_2$ has been shown to be important during the proliferation stage of zebrafish regeneration is that of hedgehog signalling[19,128,129]. A recent paper using the zebrafish tailfin regeneration model demonstrated that Sonic hedgehog (Shh) – a key signalling protein in the hedgehog signalling cascade – acted downstream of NOX to increase SOD activity and $H_2O_2$ production in the early stage of tailfin regeneration, most likely due to SOD oxidation[130]. Interestingly, although not fully elaborated on, the authors also demonstrated binding sites for HIF1α, STAT3, and NF-κB in *Shha*, suggesting that this kinase may additionally act on these factors to regulate redox balance during wound healing[130]. Importantly, Hedgehog signalling acts in close synergy to canonical Wnt signalling in regeneration as well as other pathological contexts[129,131]. In a separate study, NOX-induced $H_2O_2$ activated Wnt/β-catenin signalling and induced *FGF20* transcriptional activation to promote epidermal regeneration[18]. Canonical Wnt signalling and mitochondrial $H_2O_2$ production was also downregulated in a mouse model with *TFAM* KO, with subsequent defects in epidermal differentiation due to the inhibition of Notch signalling[132].

As well as its previously mentioned functions during the inflammation stage of wound healing, NF-κB has been shown to play significant roles during re-epithelialisation. Of note, early ROS signalling following embryonic zebrafish tailfin amputation led to the activation of NF-κB as well as promoter activation of vimentin. Here, vimentin promoted collagen formation and organisation[133]. Through serine/tyrosine phosphorylation

and ubiquitination, $H_2O_2$ acts as a regulator of IκB kinases by inducing their proteolytic degradation[134,135], which subsequently activates NF-κB and therefore promotes angiogenesis and re-epithelialisation. NF-κB is also directly regulated by $H_2O_2$-oxidation of cysteine residues in its DNA-binding region[136]. ROS has also been shown to increase the expression of TGF-β1 signalling, with downstream effects including elevated expression of collagen, fibronectin, bFGF, and matrix production[104,137]. As mitochondria are important in both TGF-β1 signalling and collagen organisation[138], as well as that of general ROS regulation during wound healing, future studies should aim to explore the potential link between these factors. Indeed, the link between ROS and TGF-β1-induced epithelial-mesenchymal transition (EMT) has been demonstrated in endometrial cell pathology[139].

$H_2O_2$ was shown to accelerate re-epithelialisation in embryonic zebrafish tailfins following wounding via the Src family kinase (SFK) FYN[140]. Although not confirmed in the particular study, FYN is known to be involved in keratinocyte differentiation[141,142]. Of note, this study confirmed findings from other studies whereby wounding led to the activation of $Ca^{2+}$, ERK, and $H_2O_2$-mediated signalling cascades independently of each other[140].

FOXO are transcription factors important in wound healing, and can either be activated by direct cysteine oxidisation, or in response to upstream redox signalling[143]. In particular, FOXO1 regulates cell cycle progression, apoptosis, and angiogenesis[144,145], and promoted both in vivo and in vitro wound healing by preventing oxidative stress in keratinocytes via the upregulation of GPX-2 and GADD45α – thus maintaining TGF-β1 mediated migration and apoptosis inhibition[146]. Additionally, low level (10 μM) $H_2O_2$ treatment induced JNK-mediated FOXO3a translocation and activation to promote AKT-mediated stem cell proliferation, which when

## Table 1 | Molecules targeting ROS in wound healing

| Category | Material | Effect | Model | Ref |
|---|---|---|---|---|
| Antioxidants | NAC | Reduces ROS levels and improves cell migration and proliferation | Cultured human gingival fibroblasts in a high-glucose environment | 172 |
| Enzymes | Glucose oxidase | Increases perfusion (via NO); collagen formation | Diabetic mice with full-thickness wounds, applied in wound dressings | 173 |
| | Superoxide dismutase | Clearance of free radicals | Hydrogels used in diabetic rat models with full-thickness wounds | 174 |
| Bio compounds | Resolvin E1 | Promotes intestinal wound repair (via CREB, mTOR, Src-FAK) | Murine biopsy-induced colonic mucosal wounds | 179 |
| | PDGF | Increases NO (higher perfusion); increased angiogenesis and cell migration | Rat model with excisional wounds, mice lacking PDGF receptors/ligands | 175,176 |
| | Galectin-1 | Effect on myofibroblast function and signalling with the release of ROS (via NOX) | Mice injected with recombinant Galectin-1 protein | 177 |
| | Alpha-arbutin | Promotes healing via upregulation of IFG1R | Cultured human dermal fibroblast | 178 |
| | Nicotinamide | Suppresses ROS; increases cell motility | Cultured human HaCaT keratinocytes | 198 |
| ROS intermediates | Topical $H_2O_2$ | Converts into available $O_2$, increase angiogenesis | Guinea pigs with ischemic wounds | 189 |
| Oxygen | Hyperbaric $O_2$ | Reduces wound hypoxia; increases fibroblast proliferation, angiogenesis and accelerated wound healing | Diabetic mouse model, in vitro model, patients (with diabetic and miscellaneous wounds) | 190–192 |
| | Topical $O_2$ | Reduces wound hypoxia; accelerates wound healing | Patients, meta-analyses | 199 |
| Nanomaterials | Copper-based nanoenzymes | ROS scavenging at low concentrations; promotes re-epithelialization and granulation | Murine diabetic model with full-thickness wounds | 183 |
| | Cerium oxide nanoparticles | ROS scavenging; stimulation of proliferation and migration of endothelial cells, keratinocytes, and fibroblasts | Human keratinocytes and microvascular endothelial cells. Mouse fibroblasts. Full-thickness wounds in mouse model | 181 |
| | PDA | Capacity of scavenging free radicals; energy transfer | PDA hydrogels in dental pulp stem cells, rat model | 182 |
| | Gold nanoparticles | Anti-inflammatory; antioxidation; enhanced wound healing | In combination with antioxidative small molecules, diabetic mouse model | 180 |
| | Selenium nanoparticles | Antibacterial; anti-inflammatory; antioxidation | In combination with hydrogels, mouse model with full-thickness wounds | 184 |
| | Carbon quantum dots | Eliminates ROS; reduces oxidative stress | Carbon dot hydrogel applied in infected wound from a mouse model | 188 |
| Others | Galvanic particles | Enhance production of ROS by keratinocytes; reduced inflammation; increased fibroblast migration | Human keratinocytes, dermal fibroblasts | 185,186 |
| | Prussian blue | Reduces ROS; increased collagen deposition; induces keratinocyte differentiation, neovascularization, and wound closure | Topically applied, mouse model | 187 |

Summary of ROS modulating materials, their effect or known mechanism of action, and the models where they were tested.
NAC N-acetyl-L-cysteine, PDGF platelet-derived growth factor, IFG1R insulin-like growth factor 1 receptor, PDA polydopamine.

transplanted, increased re-epithelialisation and angiogenesis in wounded mice[147]. Here, $H_2O_2$ promoted the production of CAT, SOD2, GPX1, and GPX2, collectively increasing stem cell proliferation and preventing oxidative stress.

Nrf2 is a cytoprotective transcription factor which is upregulated in wound healing and acts to restore redox balance by increasing levels of various antioxidants[148,149]. In physiological conditions, Nrf2 is ubiquitinated and therefore inactivated by KEAP1. However, oxidation of cysteine residues on KEAP1 – including Cys151, Cys273, and Cys288 – by $H_2O_2$ induces a conformational change in KEAP1 and stimulates the activation of Nrf2[150]. Although not studied specifically at the molecular level, the antioxidant activities of Nrf2 play important roles in wound healing – regulating apoptosis, metabolism, autophagy, angiogenesis, as well as cell proliferation and migration[151]. Various studies have attempted to utilise Nrf2-modulating compounds for chronic wound treatment[152–157]. However, more research into the specific molecular mechanisms is required in order to further elucidate the specific cellular and molecular role of Nrf2 in wound healing pathophysiology.

Finally, DUOX1-mediated $H_2O_2$ production was also shown to be important for peripheral sensory axon reinnervation in zebrafish tailfin wound healing – highlighting another import role that ROS play, even in more uncommonly studied aspects of wound healing[158].

Interestingly, by comparing ROS expression in both wound healing (early stage, lower levels) and regeneration (later stage, elevated levels) in planarians, Van Huizen et al. demonstrated that different levels of ROS act upstream of signalling pathways in a threshold-dependent manner to dictate the type of response required[122]. Similar results were found when investigating zebrafish, whereby simple wound healing required only early accumulation of ROS for JNK pathway activation, whilst injuries which necessitated new tissue formation required sustained $H_2O_2$ production, activating apoptosis pathways as well as JNK[19,130].

## Chronic wounds – redox balance abnormalities and current advancements in treatment strategies

A balanced state of oxidative stress is essential for normal wound healing. While physiological levels of ROS are required in the normal transition between wound healing phases, as previously described, an overproduction of ROS has deleterious effects and can hinder wound healing[159]. Multiple molecular mechanisms can explain this effect. For example, healing can be hindered by increased tissue damage, via opposing effects of cytokines such as VEGF and TNFα[160,161]. Excessive ROS can also alter and degrade extracellular matrix proteins and impair the function of keratinocytes and fibroblasts[162].

Diabetic wounds, another category of hard-to-heal wounds, are complex and multifactorial. Their aetiology consists classically of a triad of neuropathy, impaired vascularisation and higher susceptibility to infection. These wounds are also notoriously affected by tissue injury after prolonged hypoxia and excessive oxidative stress[93]. It is thus not surprising that targeting ROS has emerged as a potential therapy for hard-to-heal wounds, similar to other diseases, such as cancer[163,164], neurodegenerative diseases[165], T cell-mediated autoimmune diseases[166], inflammatory skin diseases[167], and others. Specifically regarding cancer, ROS play similar pleiotropic roles as they do in wound healing and chronic wound pathogenesis. Depending on the type of cancer and stage, this can include hypoxia-related ROS functions and signalling pathways such as PI3K/AKT or MAPK/ERK pathways, among others, to promote proliferation, migration, or angiogenesis. Similarly, tight regulation of ROS levels is required for cancer progression and can thus also be potentially targeted therapeutically[164].

Another factor to take into account is that of senescence, which is the phenomenon of cell cycle arrest and the inhibition of cell proliferation, and is a hallmark of several age-related pathologies, including in the pathogenesis of some chronic wounds[168]. ROS accumulation and oxidative stress can accelerate senescence in both fibroblasts[169] and endothelial cells[170], whilst UV-induced ROS upregulation additionally increases senescence and photoageing in skin[171].

Diverse strategies have emerged to modulate ROS in wound healing, namely the use of antioxidant materials such as N-acetyl-L-cysteine (NAC)[172] or enzymes which either increase local perfusion such as glucose oxidase, or clear free radicals through SODs[173,174]. Biocompounds that target ROS have also been implicated in improved wound healing, mediated by their effects on perfusion, cell migration, and ROS suppression. Examples of these molecules are Resolvin E1, PDGF, Galectin-1, Alpha-arbutin and Nicotinamide[175–179]. Nanoparticles have also been developed to improve wound healing, mainly via ROS scavenging, anti-inflammatory and anti-bacterial effects, and applied to several in vitro and in vivo models[180–184], as well as other molecules[185–188].

On the other hand, increased angiogenesis has been demonstrated when wounds were treated topically with $H_2O_2$[189] and both hyperbaric and topical oxygen led to accelerated wound healing[190–192]. The array of both pro- and anti-ROS treatment stratagies that have been applied to the study of wound healing are described in detail in Table 1.

There is, however, a lack of understanding of the exact underlying mechanisms behind the effect of these molecules in many studies and a lack of validation in human samples, which may explain the scarcity of clinical translation. It is also not always evident in the course of a chronic wound history when there is a need to reduce or stimulate ROS production, hence the importance of allying possible treatment strategies that target ROS with the use of sensors that could translate the actual needs of the wound at that given moment[193]. The utilisation of omics approaches such as scRNAseq to delineate the specific effects of ROS modulation in individual cell types during different stages of wound healing and chronic wound pathogenesis would additionally be of benefit for this purpose.

## Conclusions and perspective

ROS and their essential roles in cellular signalling regulation is one of the most important components of the complex pathophysiology of the wound healing cascade. However, owing to the multifaceted nature of wound healing and chronic wound formation, improper regulation of either ROS production or removal can lead to oxidative stress and impairment of wound healing, contributing to the formation and propagation of chronic wounds. In light of this, there has been a greater emphasis in recent years towards investigating whether ROS can be targeted to either accelerate wound healing, or conversely, be used as a treatment for chronic wounds – with no concrete advancements with regards to clinically approved therapies as of yet.

This discrepancy can partially be explained by gaps in knowledge surrounding the cellular and molecular landscape of ROS in wound healing, including for example, the role of ROS and metabolism[76,127], as well as that of redox signalling and stem cell homeostasis. In addition, future work is required in order to elucidate the specific dynamics of ROS and in particular $H_2O_2$, including the movement of $H_2O_2$ and its interplay with specific AQPs and gap junctions, or the role of ROS and other ROS-producing organelles such as the ER, which play important roles in wound healing and may potentially be targeted through pre-clinical agents[194].

Finally, it is important to note that whilst most of the investigations into the role of ROS in wound healing have been performed in animal models such as zebrafish or *Drosophila*, these studies have predominantly investigated the similar but pathophysiologically different process of regeneration, as opposed to wound healing itself[195]. However, the fact that various studies have shown that many of these pathways are activated in separate animal models[140,196,197] – such as ERK signalling – suggests that the effects of ROS are phylogenetically conserved. However, advances in methods which allow for more targeted analyses of specific cell types and their roles in different stage of wound healing pathophysiology, such as scRNAseq, are an important step in advancing knowledge in the field of ROS and wound healing.

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

## Author contributions

M.H., M.T., E.B.W. and J.W. wrote the manuscript. M.H., E.B.W., J.W. planned the manuscript. M.T. and M.H. prepared the figures.

## Funding

## Competing interests

The authors declare no competing interests.
