## [Transparent Peer Review file · Communications Biology]

Cellular and molecular roles of reactive oxygen species in wound healing

Corresponding Author: Professor Jakob Wikstrom

Version 0:

Reviewer comments:

Reviewer #1

(Remarks to the Author)
Overall impression

The manuscript by Hunt et al provides a review of the role of ROS in the process of wound healing and its pathophysiology, as well as current approaches aiming to modulate ROS to improve wound responses. The authors have attempted to cover several key areas relevant to the topic, the manuscript is overall reasonably well organised, and the diagrams used are, in principle, appropriate for enhancing the content.

The topic of the review (role of ROS in wound healing and promising ROS-modulating approaches as therapeutic interventions) is interesting and of great importance to the field of wound healing and any clinical improvements have the potential of significant impact.

However, the manuscript has substantial weaknesses, including the presence of some sections which appear 'out of place', the appropriateness of some of the references cited, whilst the figures require some improvement. Moreover, there is a large number of areas with awkward phrasing, weak sentence structure, or even confusing statements to the point where they lead to inaccuracies. Finally, improvements are required to remove grammatical issues and typos or other errors (relating to the references in particular).

Major points

1. Some of the sections appear a little out-of-place, or it has not been made clear to the reader why the material covered is relevant. In particular:

- a) For a ROS-specific review, the appearance of the nitric oxide (NO) paragraph (lines 111-115) in the "Physiological functions of ROS" section seems somewhat sudden – rather than just providing some information on NO species, would it not be worth explaining that these species are also involved in wound healing (and will be mentioned in subsequent sections)?
- b) Inclusion of the "ROS and antimicrobial activities" section (lines 172-182) is certainly relevant; yet, the importance of clearing microbial infection in the context of normal versus chronic wounds is not explicitly stated in order for this section to be linked to the topic.
- c) Lines 338-340: is the sentence "Increased angiogenesis ... wound healing" relevant in the context of the paragraph (which refers to strategies to scavenge ROS)?
- d) Mentioning the possible use of single cell RNAseq is a good point to make, yet this appears in the very last sentence of the manuscript (line 378). I believe it is a shame that the authors did not (even briefly) elaborate on the power and the potential benefit of this approach in understanding the complexities of the involvement of ROS in wound healing.

2. There are some issues with the References, whether in text or when one looks at the reference list. More specifically:

- a) Overall, a large proportion of the references of the review are not necessarily seminal relevant papers but other reviews (which is not always clear in the main text). As such, in several parts (particularly the first ~2/3 of the manuscript) the text is a 'review of reviews'.
- b) A number of citations in the reference list appear to be missing some details or have errors, e.g. refs #44, #59, #106, #135.

c) Lines 244-247: sentence "A recent paper ... SOD oxidation." – "a" paper is stated but at the end of the sentence 2 citations are provided (refs #110 and #111), so which one is the paper the authors are referring to?

d) Line 302: the authors refer to the "van Huizen et al" paper, but unless I am mistaken neither reference at the end of the sentence appears to be that paper upon checking the reference list, as van Huizen is not the first author of either ref#15 or ref#110?

3. I would pay attention to some of the statements where the phrasing is a little unfortunate and results in the statement itself being somewhat inaccurate, for instance:

a) Line 35: "concurrent" – are the wound healing phases concurrent or are they in fact "consecutive"? Strictly, they are partially overlapping, but I would not state that there are all operating concurrently.

b) Line 96: it would be more appropriate to state "antioxidant enzymes" rather than "antioxidants" as the latter is often synonymous with chemical antioxidants.

c) Line 334: the antioxidant NAC (N-acetyl-cysteine) is referred to as "antioxidant material" yet the structure of the sentence results in it being implied as an "enzyme"; please correct for accuracy.

4. In principle the figures/schematics are of good quality, however some improvements would be useful in the captions and the diagrams themselves:

a) It would be beneficial if the captions provided some narrative, so the figure can 'stand-alone' in explaining key steps, for instance in Figure 1 the cellular events denoted between phases 3 and 4.

b) In Figure 4: the schematic indicates a role for H₂O₂ in induction of Ca²⁺ uptake but the relevant text (p13, line 202) discusses the involvement of NOX4 in the activation of TRP and SERCA. It would be useful to clarify if the induction of the channels is via ROS produced by the NOXs mentioned, hence an appropriate caption could clarify that the H₂O₂ is NOX-produced (and "not shown")?

c) In Figure 5: the top left of the schematic implies NOXs and DUOXs as the sole sources of ROS (H₂O₂), but mitochondrial ROS are involved, too?

5. In parts there are examples of awkward phrasing, somewhat unclear/vague statements, and I would respectfully encourage some review of grammar usage as well as the choice of word for the purposes of narrative flow and accuracy of information, e.g.

a) Line 113: "... NO acts in an antibiotic-independent way..." – please rephrase sentence to provide clarity.

b) Lines 181-182: "... With regards to macrophages – during phagocytosis, NOX2-derived ..." – please restructure.

c) Lines 196-197: "... With regards to ... neovascularisation.", this whole sentence requires restructuring for flow and scientific clarity (please explain what is meant by "elevated NOX"; do the authors mean induced/up-regulated, over-activated...?).

d) Lines 224-225: "... several of the stimuli responsible for ... are similar with regards to re-epithelialisation" – please ensure clarity in your statement.

e) Line 276: "... although that they are separate entities" – please rephrase.

f) Line 298: "DUOX1-mediated H₂O₂" – do the authors mean 'DUOX1-induced ROS' or 'DUOX1-mediated ROS production'?

g) Lines 371-372: please correct "has" to "have" for grammatical agreement with the word "investigations".

6. Although in principle a more 'relaxed' style can make a review article a little less 'dry' and more engaging, there is a balance to be struck between that and coming across as a little too colloquial. I would improve expressions such as:

a) Line 18 "can go awry" and Line 311 heading "goes awry".

b) Line 99: I would use inverted commas for the word "sinks" as it represents an analogy.

c) Lines 151-152: "for the most part" could be rephrased to something less colloquial.

Minor points

1. Please carry out a careful check of the manuscript for typos, such as:

a) Line 47: is "milleniu" supposed to be "milieu"?

b) In Figure 1, in the schematic, the usage of colon ":" after "differentiation" (phase 3 of wound healing) is unnecessary

c) Line 175: correct "prmotes"

d) Line 205: please correct "-dependant" to "dependent".

e) Line 242: "Pi3K/" should be PI3K?

2. Other points:

a) Table 1 – only one abbreviation is provided/defined, please be consistent and either include them all or just define the abbreviation in the table itself as done for NAC for instance.

b) I note a stand-out obvious repetition in the Abstract of the word pleiotropic (lines 22 and 25), which is unfortunate and worth addressing.

Reviewer #2

(Remarks to the Author)

Recommendation: This comprehensive review by Hunt et al. effectively elucidates the molecular and cellular events

associated with ROS signaling during wound healing. The authors provide a clear summary of ROS activities, their physiological functions, cellular homeostasis mechanisms, and roles in immune responses. Additionally, the paper highlights ROS involvement in re-epithelialization and includes an extensive discussion on their cell signaling roles. The conclusion regarding ROS in chronic wounds and emerging treatments is particularly timely and relevant. I strongly recommend publication.

Minor Comments:

1. Line 82: The authors briefly touch upon how ROS spread to neighboring cells as signaling molecules, but this aspect could be further elaborated. Evidence suggests that ROS can diffuse through aquaporins, gap junctions, and extracellularly. Specifically, gap junctions facilitate the intercellular spread of ROS, which may contribute to the propagation of oxidative stress in tissues. Adding a sentence on this topic would enhance the manuscript's clarity. Relevant references include (any of those or others, can help):

doi: 10.1089/ars.2008.2146

<https://www.ncbi.nlm.nih.gov/pmc/articles/PMC2763361/>

<https://www.mdpi.com/2076-3921/10/9/1374>

DOI:10.1371/journal.pone.0041633

2. Line 175: The word "promotes" should be checked.

3. Line 241: Ensure "thioredoxin" is corrected.

4. Line 249: Confirm that "Shh" is appropriately introduced.

5. AKT Signaling: The discussion on AKT's role in the paper is insightful. It may be beneficial to mention that ROS can influence the insulin signaling pathway, which in turn affects AKT activity. Research has indicated that nutrients promoting insulin signaling facilitate AKT-mediated regenerative growth in a ROS-dependent manner (doi: 10.1242/dev.197087). Including this connection would strengthen the discussion on ROS's regulatory roles in metabolic pathways during wound healing.

6. Line 105 check the sentence "and can be"

7. Line 160 Ensure "MAO" is corrected.

8. Line 265 and 267. "TGF β 1" should be corrected

9. It would be nice to comment ROS in cancer (Chronic Wound section, after the second paragraph).

Version 1:

Reviewer comments:

Reviewer #1

(Remarks to the Author)

I am happy with the revised manuscript and I believe that the authors have addressed all my points of critique adequately. I am happy to recommend acceptance for publication.

Reviewers' comments:

Reviewer #1 (Remarks to the Author):

Overall impression

The manuscript by Hunt et al provides a review of the role of ROS in the process of wound healing and its pathophysiology, as well as current approaches aiming to modulate ROS to improve wound responses. The authors have attempted to cover several key areas relevant to the topic, the manuscript is overall reasonably well organised, and the diagrams used are, in principle, appropriate for enhancing the content.

The topic of the review (role of ROS in wound healing and promising ROS-modulating approaches as therapeutic interventions) is interesting and of great importance to the field of wound healing and any clinical improvements have the potential of significant impact.

However, the manuscript has substantial weaknesses, including the presence of some sections which appear 'out of place', the appropriateness of some of the references cited, whilst the figures require some improvement. Moreover, there is a large number of areas with awkward phrasing, weak sentence structure, or even confusing statements to the point where they lead to inaccuracies. Finally, improvements are required to remove grammatical issues and typos or other errors (relating to the references in particular).

Major points

1. Some of the sections appear a little out-of-place, or it has not been made clear to the reader why the material covered is relevant. In particular:

a) For a ROS-specific review, the appearance of the nitric oxide (NO) paragraph (lines 111-115) in the "Physiological functions of ROS" section seems somewhat sudden – rather than just providing some information on NO species, would it not be worth explaining that these species are also involved in wound healing (and will be mentioned in subsequent sections)?

Thank you for taking the time to review our paper as well as for providing several useful suggestions. For this particular point, we have expanded on the role of NO in wound healing (lines 125-135).

b) Inclusion of the "ROS and antimicrobial activities" section (lines 172-182) is certainly relevant; yet, the importance of clearing microbial infection in the context of normal versus chronic wounds is not explicitly stated in order for this section to be linked to the topic.

We agree with the point raised, and have now added more detail about the importance of clearing microbial infection to prevent chronic wounds (lines 211-214).

c) Lines 338-340: is the sentence "Increased angiogenesis ... wound healing" relevant in the context of the paragraph (which refers to strategies to scavenge ROS)?

Thank you for raising this point, we agree that this is slightly confusing. As we talk about both pro and anti ROS strategies we have re-ordered the text to make clear the distinction (lines 396-408).

d) Mentioning the possible use of single cell RNAseq is a good point to make, yet this appears in the very last sentence of the manuscript (line 378). I believe it is a shame that the authors did not (even briefly) elaborate on the power and the potential benefit of this approach in understanding the complexities of the involvement of ROS in wound healing. We agree with this point, and so have now added text about the potential of scRNAseq to ROS in wound healing studies in several places (lines 199-203; 414-417; 447).

2. There are some issues with the References, whether in text or when one looks at the reference list. More specifically:

a) Overall, a large proportion of the references of the review are not necessarily seminal relevant papers but other reviews (which is not always clear in the main text). As such, in several parts (particularly the first ~2/3 of the manuscript) the text is a 'review of reviews'. We thank the reviewer for pointing this out. We have now updated the manuscript to include seminal relevant papers as opposed to just review papers.

b) A number of citations in the reference list appear to be missing some details or have errors, e.g. refs #44, #59, #106, #135.

Thank you for point this out. The references have now been updated to eliminate errors.

c) Lines 244-247: sentence "A recent paper ... SOD oxidation." – "a" paper is stated but at the end of the sentence 2 citations are provided (refs #110 and #111), so which one is the paper the authors are referring to?

Thank you for raising this point. The correct reference was the Thauvin et al. paper. We have removed the other reference.

d) Line 302: the authors refer to the "van Huizen et al" paper, but unless I am mistaken neither reference at the end of the sentence appears to be that paper upon checking the reference list, as van Huizen is not the first author of either ref#15 or ref#110?

With regards to this point, the Van Huizen paper is referenced at the beginning of the paragraph, whereas the other two references are from papers which describe similar results in zebrafish.

3. I would pay attention to some of the statements where the phrasing is a little unfortunate and results in the statement itself being somewhat inaccurate, for instance:

a) Line 35: "concurrent" – are the wound healing phases concurrent or are they in fact

"consecutive"? Strictly, they are partially overlapping, but I would not state that there are all operating concurrently.

Thank you for raising this point. We have rephrased this sentence to make it clear that it is "consecutive" (line 34).

b) Line 96: it would be more appropriate to state "antioxidant enzymes" rather than "antioxidants" as the latter is often synonymous with chemical antioxidants.

We agree with the point raised and have changed it to "antioxidant enzymes" (line 108).

c) Line 334: the antioxidant NAC (N-acetyl-cysteine) is referred to as "antioxidant material" yet the structure of the sentence results in it being implied as an "enzyme"; please correct for accuracy.

Thank you for raising this point. We have now altered the text to clarify that NAC is an antioxidant material (line 397).

4. In principle the figures/schematics are of good quality, however some improvements would be useful in the captions and the diagrams themselves:

a) It would be beneficial if the captions provided some narrative, so the figure can 'stand-alone' in explaining key steps, for instance in Figure 1 the cellular events denoted between phases 3 and 4.

Thank you for raising this point. We agree, and have now expanded the descriptions in each of the figure legends to make them more narrative.

b) In Figure 4: the schematic indicates a role for H₂O₂ in induction of Ca²⁺ uptake but the relevant text (p13, line 202) discusses the involvement of NOX4 in the activation of TRP and SERCA. It would be useful to clarify if the induction of the channels is via ROS produced by the NOXs mentioned, hence an appropriate caption could clarify that the H₂O₂ is NOX-produced (and "not shown")?

We agree with the point raised, and as such have updated the figure now to clarify that the H₂O₂ is NOX4 produced.

c) In Figure 5: the top left of the schematic implies NOXs and DUOXs as the sole sources of ROS (H₂O₂), but mitochondrial ROS are involved, too?

This is a valid point raised. We have now updated the figure and figure legend to also indicate that mitochondria are also a source of H₂O₂

5. In parts there are examples of awkward phrasing, somewhat unclear/vague statements, and I would respectfully encourage some review of grammar usage as well as the choice of word for the purposes of narrative flow and accuracy of information, e.g.

a) Line 113: "... NO acts in an antibiotic-independent way..." – please rephrase sentence to provide clarity.

Thank you for pointing this out. As this sentence was unnecessary and unclear, we have now deleted it (lines 209-214).

b) Lines 181-182: "... With regards to macrophages – during phagocytosis, NOX2-derived ..." – please restructure.

Thank you for the suggestion. We have now restructured this sentence to make it clearer (lines 220-222).

c) Lines 196-197: "... With regards to ... neovascularisation.", this whole sentence requires restructuring for flow and scientific clarity (please explain what is meant by "elevated NOX"; do the authors mean induced/up-regulated, over-activated...?).

Thank you for raising this point. We have now restructured this sentence and specified that it is elevated NOX activation that leads to increased levels of NO (lines 241-243).

d) Lines 224-225: "... several of the stimuli responsible for ... are similar with regards to re-epithelialisation" – please ensure clarity in your statement.

We agree with this suggestion and have thus restructured the sentence to provide more clarity on the statement (lines 272-273).

e) Line 276: "... although that they are separate entities" – please rephrase.

Thank you for this suggestion. We have now rephrased this sentence to provide more clarity (lines 330-331).

f) Line 298: "DUOX1-mediated H₂O₂" – do the authors mean 'DUOX1-induced ROS' or 'DUOX1-mediated ROS production'?

Thank you for raising this point. We have changed the sentence to state DUOX1-mediated H₂O₂ production (line 353).

g) Lines 371-372: please correct "has" to "have" for grammatical agreement with the word "investigations".

Thank you for pointing this mistake out. We have now changed this (line 441).

6. Although in principle a more 'relaxed' style can make a review article a little less 'dry' and more engaging, there is a balance to be struck between that and coming across as a little too colloquial. I would improve expressions such as:

a) Line 18 "can go awry" and Line 311 heading "goes awry".

Thank you for your suggestions. We agree with the point(s) raised and have changed the expression in both instances (line 18; line 370).

b) Line 99: I would use inverted commas for the word "sinks" as it represents an analogy.

We have updated this now to use inverted commas (line 111).

c) Lines 151-152: "for the most part" could be rephrased to something less colloquial.

We have changed the phrasing now to follow this suggestion (line 180).

Minor points

1. Please carry out a careful check of the manuscript for typos, such as:

a) Line 47: is "milleniu" supposed to be "milieu"?

Yes, we agree with the point raised and have now corrected this mistake (line 46).

b) In Figure 1, in the schematic, the usage of colon ":" after "differentiation" (phase 3 of wound healing) is unnecessary

Thank you for this suggestion. We have updated figure 1 to include this change.

c) Line 175: correct "prmotes"

Thank you for noticing this mistake. We have now updated it with the correct spelling (line 210).

d) Line 205: please correct "-dependant" to "dependent."

We have changed this to correct the mistake (line 260).

e) Line 242: "Pi3K/" should be PI3K?

Yes, we agree with the reviewer and have now updated this (line 388).

2. Other points:

a) Table 1 – only one abbreviation is provided/defined, please be consistent and either include them all or just define the abbreviation in the table itself as done for NAC for instance.

We agree with the point raised and have now defined abbreviations in the table legend.

b) I note a stand-out obvious repetition in the Abstract of the word pleiotropic (lines 22 and 25), which is unfortunate and worth addressing.

We also agree with this point, and have removed one instance of pleiotropic.

Reviewer #2 (Remarks to the Author):

Recommendation: This comprehensive review by Hunt et al. effectively elucidates the molecular and cellular events associated with ROS signaling during wound healing. The authors provide a clear summary of ROS activities, their physiological functions, cellular homeostasis mechanisms, and roles in immune responses. Additionally, the paper highlights ROS involvement in re-epithelialization and includes an extensive discussion on their cell signaling roles. The conclusion regarding ROS in chronic wounds and emerging treatments is particularly timely and relevant. I strongly recommend publication.

Minor Comments:

1. Line 82: The authors briefly touch upon how ROS spread to neighboring cells as signaling molecules, but this aspect could be further elaborated. Evidence suggests that ROS can diffuse through aquaporins, gap junctions, and extracellularly. Specifically, gap junctions facilitate the intercellular spread of ROS, which may contribute to the propagation of oxidative stress in tissues. Adding a sentence on this topic would enhance the manuscript's clarity. Relevant references include (any of those or others, can help):
doi: 10.1089/ars.2008.2146

<https://www.ncbi.nlm.nih.gov/pmc/articles/PMC2763361/>

<https://www.mdpi.com/2076-3921/10/9/1374>

DOI:10.1371/journal.pone.0041633

Thank you for taking the time to review of paper, as well as for providing useful suggestions and references. With regards to this point, we agree with the importance of mentioning gap junctions, and have now included text to this end (lines 91-93).

2. Line 175: The word "promotes" should be checked.

Thank you for point this out. We have corrected this now (line 210).

3. Line 241: Ensure "thioredoxin" is corrected.

Thank you for noticing this mistake. We have corrected this now (line 289).

4. Line 249: Confirm that "Shh" is appropriately introduced.

We agree with this point raised and have included addition text to introduce Shh more appropriately (line 300-301).

5. AKT Signaling: The discussion on AKT's role in the paper is insightful. It may be beneficial to mention that ROS can influence the insulin signaling pathway, which in turn affects AKT activity. Research has indicated that nutrients promoting insulin signaling facilitate AKT-mediated regenerative growth in a ROS-dependent manner (doi: 10.1242/dev.197087). Including this connection would strengthen the discussion on ROS's regulatory roles in metabolic pathways during wound healing.

We agree with and are appreciative of the reviewers comment here, and have thus included addition text to expand on this point (lines 289-297).

6. Line 105 check the sentence "and can be"

Thank you for making us aware of this mistake. We have adjusted accordingly.

7. Line 160 Ensure "MAO" is corrected.

Thank you for pointing this out. We have corrected this mistake (line 188).

8. Line 265 and 267. "TGFB1" should be corrected

Thank you for making us aware of these typos. We have checked and corrected throughout.

9. It would be nice to comment ROS in cancer (Chronic Wound section, after the second paragraph).

We agree with this suggestion and have now added additional text to this end (lines 385-390).